# Estimation of Temporal Muon Signals in Water-Cherenkov Detectors of the Surface Detector of the Pierre Auger Observatory

Margita Kubátová [1,2], for the Pierre Auger Collaboration[3]⋆

**1** Institute of Physics of the Czech Academy of Sciences, Prague, Czechia
**2** Czech Technical University in Prague, FNSPE, Prague, Czechia
**3** Observatorio Pierre Auger, Av. San Martín Norte 304, 5613 Malargüe, Argentina
Full author list: https://www.auger.org/archive/authors_2025_08.html

⋆ spokespersons@auger.org

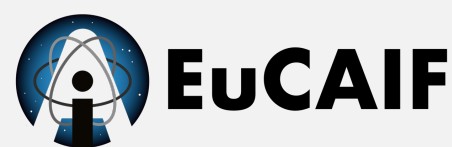

*The 2nd European AI for Fundamental Physics Conference (EuCAIFCon2025) Cagliari, Sardinia, 16-20 June 2025*

## Abstract

**The Surface Detector (SD) of the Pierre Auger Observatory is a 3000 km² array of stations, whose main components are Water-Cherenkov Detectors (WCDs) recording ground-level signals from extensive air showers (EASs) initiated by Ultra-High-Energy Cosmic Rays (UHECRs). Understanding the physics of UHECRs requires knowledge of their mass composition, for which the number of ground muons is a key probe. Isolating the muon component is difficult, as different types of particles contribute to the SD signal. We apply a recurrent neural network to estimate the muon content of the SD signals, showing small bias in simulations and weak dependence on selected hadronic interaction model.**

## 1 Introduction

Ultra-High-Energy Cosmic Rays (UHECRs) are charged extraterrestrial particles of Galactic and extragalactic origin with energies above $10^{18}$ eV. Identifying their sources and understanding the mechanisms that accelerate particles to such extreme energies remain central challenges in astroparticle physics. Determining the mass composition of UHECRs is crucial to identify their sources, as their deflection by galactic and intergalactic magnetic fields during their journey to Earth depends on their energy-to-charge ratio, and thus on their composition.

When a UHECR enters the atmosphere, it initiates a cascade of secondary particles that propagate towards the ground, forming an Extensive Air Shower (EAS). The Pierre Auger Observatory has a central role in studying these events [1] due to its high exposure. The main detectors of the Observatory are the Fluorescence Detector (FD), observing the longitudinal development of the EAS in the atmosphere, and the Surface Detector (SD), an array of 1,660

Water-Cherenkov Detectors (WCDs) covering 3000 km$^2$, which record the spatial and temporal distribution of particles at ground level via Cherenkov light produced by relativistic charged particles in water. The core part of the SD is the SD-1500, an array with 1500 m spacing between stations, fully efficient for energies above $10^{18.5}$ eV. Two nested denser arrays with 750 m (SD-750) and 433 m (SD-433) spacing extend the detection to lower energies.

While the FD offers a calorimetric measurement, its duty cycle is only about 15% of the time. In contrast, the SD operates nearly continuously, making it crucial for mass composition studies at the highest energies, where event statistics are limited.

## 1.1  Extracting the muon signal from SD

The WCD signal is a superposition of contributions from muons, electrons, and photons, with a minor hadronic component. Separating these components is essential for mass composition analyses, as the number of muons on the ground correlates with the mass of the primary cosmic ray – EAS induced by heavier nuclei produces stochastically more muons. Each signal is digitized with a sampling rate of 40 MHz as a time series (the trace) with 768 bins, capturing the temporal structure of particle arrivals. Different particle types leave distinct signatures in the trace: muons, being less scattered in the atmosphere, tend to arrive earlier and produce sharp, isolated peaks, while electrons and photons generate broader, smoother pulses.

Building on the approach developed in [2], which demonstrated that these differences can be effectively exploited using Recurrent Neural Networks (RNNs) to identify the muonic component, we extend the method using updated simulations, broader input data, and optimized NN architecture.

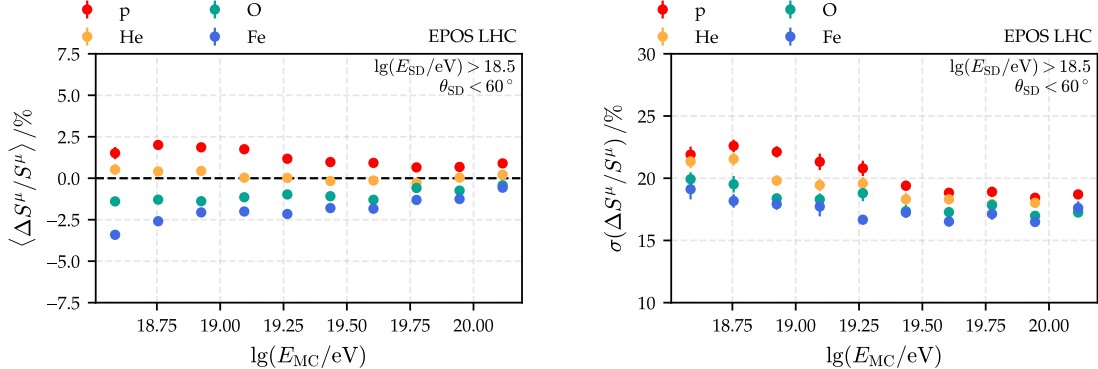

Figure 1: Relative bias (left) and resolution (right) of the muon signal $S^\mu$ as a function of Monte Carlo simulated energy for EAS induced by hydrogen (p, red), helium (He, yellow), oxygen (O, green), and iron (Fe, blue) nuclei.

## 2  The neural network

The model is trained and evaluated using events from the Auger reference air shower library [3], simulated using CORSIKA 7.7420 [4]. For training, we used EPOS LHC [5] as the default hadronic interaction model. In addition, we tested the NN on the hadronic interaction model Sibyll$^\star$ [6]. The library includes four primary particle types — proton, helium, oxygen, and iron — with energies between $10^{18.5}$ and $10^{20.2}$ eV. Showers follow an $E^{-1}$ energy spectrum and a zenith-angle distribution uniform in $\sin^2\theta$ up to 65°. The detector responses were simulated, and the event reconstruction was performed using the Auger $\overline{\text{Off}\underline{\text{line}}}$ framework.

A key improvement over [2] came from incorporating low gain (LG) traces. Each WCD provides output in two channels with different amplification: high gain (HG) for lower sig-

nals, typically from distant stations, and LG for larger signals, closer to the shower core. By incorporating both HG and LG traces, the network can reconstruct the muon content across the full dynamic range of the detector.

The network input consists of two parts. The first is a set of three scalar parameters — the shower zenith angle $\theta$, the azimuth angle $\zeta$ of the station in the shower plane, and the distance from the shower core $r$ — which encode information about the amount of atmosphere traversed by the particles before reaching the detector. The second part consists of the first 200 bins of the WCD trace after the trigger. Using the full 768 bins degrades the LSTM performance due to limited memory. Restricting the input to the first 200 bins mitigates this issue while preserving sufficient information, as the muon signal is concentrated mainly in the early part of the trace. The scalar inputs are processed by dense layers and used as initial values of the hidden and cell states of the LSTM layers, which then take the trace as input. The LSTM output is passed through additional dense layers to predict the muon trace — the muon content in each of the 200 time bins. The model was implemented in Keras [7] and optimized using Keras Tuner [8]. Training was performed over 300 epochs with a batch size of 512, using the Adam optimizer [9] with a learning rate of $10^{-4}$ and mean squared error as the loss function.

Performance is evaluated by comparing the sum of the predicted and true muon traces, denoted $\widehat{S^{\mu}}$ and $S^{\mu}$, respectively, with the difference defined as $\Delta S^{\mu} = \widehat{S^{\mu}} - S^{\mu}$.

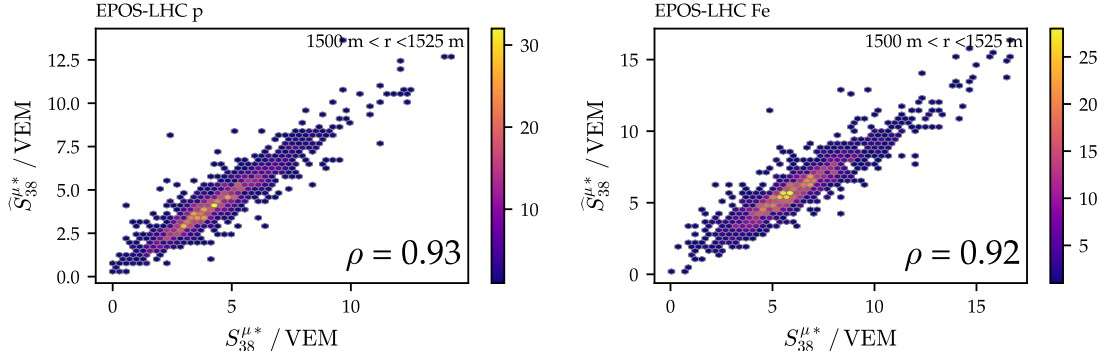

Figure 2: Correlation of true and predicted muon signals, normalized to a reference energy and zenith angle, for proton (left) and iron (right). Each plot shows data within a narrow distance range from the shower core, as indicated in the top part of the figure. Color denotes the number of stations with corresponding muon signal.

## 2.1 Performance on simulated events

For energies $\lg(E/\text{eV}) \gtrsim 18.7$, the network achieves a relative bias below $\pm 2.5\%$ (Fig. 1).

Fig. 2 shows the strong correlation between the normalized true muon signal $S_{38}^{\mu*}$ and the predicted $\widehat{S}_{38}^{\mu*}$. The asterisk denotes normalization to an energy of $10^{19}$ eV, and the subscript 38 to a zenith angle of $38°$. This normalization, together with restricting the analysis to a narrow core-distance range, reduces correlations with $(E, r, \theta)$.

Two example traces in Fig. 3 demonstrate the network's ability to recover the fast, narrow peaks associated with muon hits.

## 2.2 Dependence on hadronic interaction models

To assess the robustness of the method, the NN trained on EPOS LHC simulations was evaluated on simulations of Sibyll$^{\star}$, which has $\sim 30\%$ more muons at ground level. Even though the NN was not trained on Sibyll$^{\star}$, the bias and resolution remain similar in magnitude, with only a small global shift, as shown in Fig. 4. This weak dependence on the hadronic interaction

model is an important feature for applications to real data, where the true underlying physics is unknown.

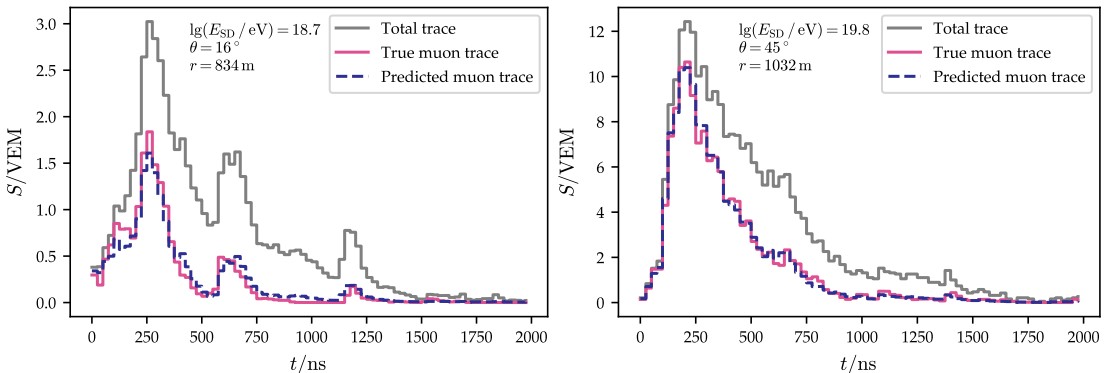

Figure 3: Examples of true and predicted muon traces in simulations.

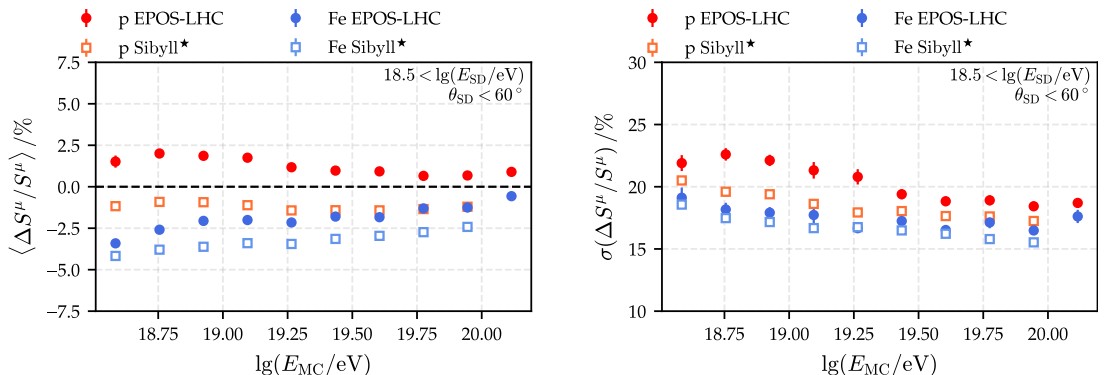

Figure 4: Relative bias (left) and resolution (right) of the muon signal $S^\mu$ versus Monte Carlo energy for EAS induced by hydrogen (p) and iron (Fe). Results for the NN trained on EPOS LHC (red, blue) and evaluated on Sibyll$^\star$ (orange, sky blue).

## 3  Conclusion

We demonstrated that the muon component of SD signals can be effectively extracted using a model based on RNNs. The model exhibits a bias below ±2.5% across a broad energy range and generalizes well to events simulated with different hadronic interaction models, supporting its applicability to real data.

An important validation of the network performance may be achieved with data from the Underground Muon Detector (UMD), a dedicated muon counter buried in the SD-750, which provides a direct measurement of air shower muons. Further developments will leverage the improved electronics and additional components of AugerPrime [10], including surface scintillator detectors, to enable event-by-event mass composition studies at the highest energies.

## Acknowledgements

**Funding information**  This work was co-funded by the European Union and supported by the Czech Ministry of Education, Youth and Sports (Project No. FORTE – CZ.02.01.01/00/22 008/0004632) and GACR (Project No. 24-13049S).

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
