# Peer review of "Estimation of Temporal Muon Signals in Water-Cherenkov Detectors of the Surface Detector of the Pierre Auger Observatory"

_SciPost Physics Proceedings_

## Round 1 · Referee Report · Anonymous (Referee 1) · 2025-12-1

Report

The proceedings article reports on the results of a study that leverages recurrent neural networks to identify the muon component in the surface detector signals of the Pierre Auger Observatory in order to enhance the sensitivity of the telescope to the mass composition of the incident primary ultra-high-energy cosmic rays.

I find the presentation of the material succinct and descriptive enough to understand the approach of the authors. The findings demonstrate the viability of the applied machine-learning method in the context of Pierre Auger data. In this regard, I support the publication of this proceedings article. Yet, I have very minor comments that the authors may address.

Requested changes

  1. In Section 2, I read the acronym "LSTM", which seems to be undefined.
  2. In the left panel of Fig. 4, the points for all tested models seem to be either systematically above or below the 0% line without crossing between negative and positive deviations. Is this some systematic bias that one should expect or an artefact of the neural network?
  3. In the conclusions, it might be interesting for the reader to find a few other sources of uncertainty that one might need to address once real data is being analysed with this method.

Recommendation

Publish (meets expectations and criteria for this Journal)

  • validity: high
  • significance: good
  • originality: good
  • clarity: good
  • formatting: excellent
  • grammar: excellent

Author:  Margita Kubátová  on 2025-12-15  [id 6147]

(in reply to Report 1 on 2025-12-01)

Thank you for the positive review, the responses to the individual points are provided below: 1. The missing LSTM definition has been added. 2. The plot shows the bias for proton and iron induced showers for two hadronic interaction models. Since proton showers contain fewer muons than iron showers at the same energy, predictions from the NN tend to fall between the two cases. As a result, some overprediction for protons and underprediction for irons is expected. Thus, this behaviour reflects the underlying shower physics rather than an artefact of the network architecture. 3. Work on systematics is currently in the progress. Due to the limited length of the proceedings, we only provided a relevant source for readers interested in potential sources of corrections.

---

## Editorial Decision

editorial_decision: